# Laccase Mediator Cocktail System as a Sustainable Skin Whitening Agent for Deep Eumelanin Decolorization

**DOI:** 10.3390/ijms23116238

**Published:** 2022-06-02

**Authors:** Valeria Gigli, Davide Piccinino, Daniele Avitabile, Riccarda Antiochia, Eliana Capecchi, Raffaele Saladino

**Affiliations:** 1Department of Experimental Medicine, Sapienza University of Rome, Viale Regina Elena 324, 00166 Rome, Italy; valeria.gigli@uniroma1.it; 2Department of Biological and Ecological Sciences, University of Tuscia, 01100 Viterbo, Italy; d.piccinino@unitus.it; 3IDI Farmaceutici, Pomezia, 00071 Rome, Italy; davitabile@idifarmaceutici.it; 4Department of Chemistry and Drug Technologies, Sapienza University of Rome, P.le Aldo Moro 5, 00185 Rome, Italy; riccarda.antiochia@uniroma1.it

**Keywords:** laccase mediator cocktail system, skin whitening agents, eumelanin decolorization

## Abstract

The overproduction of eumelanin leads to a panel of unaesthetic hyper-pigmented skin diseases, including melasma and age spots. The treatment of these diseases often requires the use of tyrosinase inhibitors, which act as skin whitening agents by inhibiting the synthesis of eumelanin, with harmful side effects. We report here that laccase from *Trametes versicolor* in association with a cocktail of natural phenol redox mediators efficiently degraded eumelanin from *Sepia officinalis*, offering an alternative procedure to traditional whitening agents. Redox mediators showed a synergistic effect with respect to their single-mediator counterpart, highlighting the beneficial role of the cocktail system. The pro-oxidant DHICA sub-units of eumelanin were degraded better than the DHI counterpart, as monitored by the formation of pyrrole-2,3,5-tricarboxylic acid (PTCA) and pyrrole-2,3-dicarboxylic acid (PDCA) degradation products. The most effective laccase-mediated cocktail system was successively applied in a two-component prototype of a topical whitening cream, showing high degradative efficacy against eumelanin.

## 1. Introduction

The overproduction of eumelanin by melanocytes causes skin hyperpigmentation disorders, such as melasma as well as age and post-inflammatory spots, which can affect human health and social relationships [1,2]. Current methods for skin whitening are aggressive and harmful to a person’s health [3,4]. Most of the commercial skin whitening agents are inhibitors of melanogenesis at the cellular level, favoring skin turnover by melanosome dispersion [5]. Among them, arbutin, aloesin, kojic acid, α-lipoic acid, hydroquinone, and corticosteroids are characterized by side effects, including dermatitis, acne, allergy, neuropathies, hypertension, and kidney damage [6,7,8,9,10,11,12,13]. Oxidative enzymes that are able to degrade melanin, such as peroxidase (EC 1.11.1.7) and laccase (EC 1.10.3.2), have received great interest as a topical alternative to common tyrosinase inhibitors [14,15]. The use of peroxidase requires H_2_O_2_ as a primary oxidant, which favors skin irritation [16,17,18,19,20,21,22,23,24,25]. On the contrary, laccase is able to activate dioxygen as a primary oxidant without the necessity for peroxide intermediates [26]. In this latter case, low-molecular-weight redox mediators are required for the degradation of melanin [27]. These compounds perform as redox shuttles from the active site of the enzyme to the bulk of the solution, thus improving the overall reactivity (laccase mediator system, LMS). In the absence of redox mediators, the treatment with laccase increases the complexity of melanin due to the occurrence of oxidative radical coupling processes [26]. Examples of the LMS degradation of melanin have been reported, as well as the use of a single redox mediator such as 1-hydroxybenzotriazole (HBT), 2,2′-azino-bis(3-ethylbenzothiazoline-6-sulfonic acid) (ABTS), acetosyringone, syringaldehyde, *p*-coumaric acid, vanillin, vanillic acid, vanillyl alcohol, and acetovanillone, which are discussed in detail in [28,29,30,31]. Irrespective of the experimental conditions, these processes are affected by a low degree of efficacy as well as by the use of high-cost enzymes and long reaction times. We report here that a laccase mediator cocktail system (LMCS) improves the degradation of melanin from *Sepia officinalis,* a well-recognized human eumelanin model, with respect to a single laccase mediator [32,33,34]. We used laccase from *Trametes versicolor,* a low-cost and commercially available enzyme, and a pair of redox mediators—ABTS and (2,2,6,6-tetramethylpiperidin-1-yl)oxyl (TEMPO); alternatively, we also used natural phenols such as vanillin (V), syringaldehyde (Syr), acetosyringone (As), acetovanillone (Av), and vanillyl alcohol (Va). These compounds covered a wide range of possible oxidative reaction mechanisms [35,36,37]. Eumelanin was degraded efficiently (up to 87%), the best results being obtained with the pair, V/Va. The residual eumelanin was characterized by scanning electron microscopy (SEM) and Fourier Transform Infrared spectroscopy (FT-IR), showing a high degradation of the polymer with the disappearance of typical spherical nanostructures. The degradation was further increased to 96% by a second run of the treatment. The formation of degradation markers, pyrrole-2,3,5-tricarboxylic acid (PTCA) and pyrrole-2,3-dicarboxylic acid (PDCA), provided useful information about the reaction pathway [38,39,40,41,42,43,44]. The optimal LMCS was applied in the preparation of a prototype of a topical whitening cream, characterized by a degradative efficacy of up to 78%.

## 2. Results

### 2.1. Decolorization of Eumelanin from Sepia Officinalis by Laccase Mediator System (LMS)

Initially, we studied the degradation of eumelanin from *Sepia officinalis* by the traditional LMS using a single redox mediator [26]. The experiments were performed using TEMPO and ABTS as synthetic compounds in comparison with five natural redox mediators from lignocellulose wastes: Syr, V, As, Av, and Va [45]. The selected mediators were characterized by three alternative redox mechanisms, namely electron transfer ET (ABTS) [35], ionic IM (TEMPO) [36], and hydrogen atom transfer HAT (Syr, V, As, Av, and Va) mechanisms [37]. As a general procedure, eumelanin (800 μg) and three different amounts of the selected redox mediator (0.4 μmol, 0.8 μmol, and 1.6 μmol) in NaOH (200 μL, 0.5 M) and citrate buffer (2.4 mL; 0.1 M, pH 6) were treated with laccase (0.79 U/mg) in citrate buffer (pH 6.0, 400 µL) at 37 °C for 4 h. The degradation process was monitored by measuring the absorbance of the sample at ʎ_MAX_ of eumelanin (540 nm) [46], while the degradation efficiency DE (Table 1) was evaluated by applying the following equation [16]: (1)$$\left(DE\right)\;Decolorization\;Efficency\;\%=\frac{Abs_{0\left(540\;\mathrm{nm}\right)}-Abs_{f\left(540\;\mathrm{nm}\right)}}{Abs_{0\;\left(540\;\mathrm{nm}\right)}}\times 100$$
where *Abs*_0_ is the initial value of absorbance, and *Abs_f_* is the value of absorbance after 4 h. The treatment of eumelanin in the absence of the redox mediator was performed as a reference [30]. In this latter case, the absorbance of the system increased with time (Appendix A), suggesting the occurrence of the expected oxidative cross-coupling processes [26].

The highest value of DE was obtained in the presence of V (Table 1, entry 3), followed by Va and ABTS (Table 1, entries 7 and 1, respectively). As a general trend, the value of DE increased by increasing the amount of the redox mediator, with the only exceptions showed by Syr and TEMPO. These data suggest that the degradation and oligomerization pathways were both involved in the process, depending on the nature of the redox mediator. In particular, the HAT (associated with V and Va) and ET (ABTS) mechanisms were the most efficient processes in the degradation of eumelanin under LMS conditions. These results are of particular relevance since V and Va are natural phenols from sustainable raw materials [47]. Compounds V and Va were previously reported as redox mediators in the degradation of synthetic melanin with laccase from *Lentinus polychrous* [28]. 

### 2.2. Decolorization of Eumelanin by Laccase Mediator Cocktail System (LMCS)

The laccase mediator cocktail system (LMCS) shows the contemporary presence of (at least) two redox mediators, sometimes involving different reaction mechanisms. In nature, LMCS is involved in the degradation of lignin by fungi through the cooperative action of lytic polysaccharide monooxygenase and oligomers deriving from lignin degradation [34,48,49]. In order to improve the efficacy of the degradation of eumelanin, we studied the use of LMCS by combining the following pairs of redox mediators at different ratios (0.5:1; 1:1; 2:1) in order to reach a total amount of 1.6 μmol: Va/V, ABTS/V, Syr/V, TEMPO/V, As/V, Av/V, As/Av, Av/Syr, and As/Syr. As a general procedure, eumelanin (800 μg) and a selected pair of redox mediators (1.6 μmol) in NaOH (200 μL, 0.5 M) and citrate buffer (2.4 mL; 0.1 M, pH 6) were treated with laccase (0.79 U/mg) in citrate buffer (pH 6.0, 400 µL) at 37 °C for 4 h. The degradation of eumelanin was monitored by measuring the absorbance of the system at the maximum melanin ʎ_MAX_ of 540 nm [46]. Irrespective of the experimental conditions, the highest value of DE was always obtained using similar equivalents of redox mediators (ratio 1:1) (Table 2). The Va/V couple afforded the highest DE value (67%), followed by Syr/V (55%), ABTS/V (50%), TEMPO/V (46%), and As/V (39%) (Table 2, entries 17, 8, 2, 5, and 11, respectively). Note that vanillin was a common ingredient in the most effective LMCS, suggesting the major role of this compound in the degradation of eumelanin. In addition, the Va/V and Syr/V couples showed a synergistic effect. The LMCS treatment of eumelanin at the optimal ratio of redox mediators (1:1 ratio) was repeated at 24 h and 48 h (Table 2, left column). At 24 h of treatment, the highest value of DE (87%) was obtained in the presence of Va/V, confirming the efficacy of this system in the degradation of eumelanin. A longer reaction time (48 h) did not noticeably increase the DE value (the color intensity of eumelanin after treatment with LMCS at 24 h is reported in Figure 1). In the case of the five most active couples, Va/V, ABTS/V, Syr/V, TEMPO/V, and As/V (1:1 ratio), we afforded a second cycle of treatment of eumelanin at 24 h (Table 2). It is worth noting that laccase in the presence of Va/V quantitatively degraded eumelanin after the second run.

### 2.3. Analysis of LMCS Oxidative Pathway

In order to gain information about the degradation pathway of eumelanin by LMCS, we analyzed the sample after its treatment with the five best pairs of redox mediators by gas chromatography associated with mass spectrometry (GC-MS), quantifying the formation of PTCA and PDCA, which are considered the main degradation products of eumelanin. The formation of these compounds is indicative of the degradation of specific melanin sub-units, PTCA being correlated with the degradation of DHICA, whilst PDCA with that of DHI [50,51]. The interaction between laccase and eumelanin produces a phenoxyl radical at the DHICA and DHI sub-units through the abstraction of one electron from the substrate, followed by oxidation and the successive ring opening of benzoquinone intermediates to yield PTCA and PDCA [38]. As a general procedure, the residual eumelanin recovered after the treatment with the five best LMCS was precipitated by adding HCl (2.2 mL, 6.0 M) and the recovered solution extracted with ethyl acetate (20 mL; ×3) and evaporated under reduced vacuum. The residue was treated with bis-trimethyl silyl trifluoro acetamide (BSTFA, 300 μL) in the presence of *n*-dodecane (7.8 μmol) as an internal standard and analyzed by GC-MS. PTCA and PDCA were identified at 19 min and 21 min, respectively. Mass fragmentation analysis confirmed the structure of these compounds [52]. Note that PTCA and PDCA were absent in the original sample of eumelanin. Irrespective of the experimental conditions, PTCA was detected in higher amounts than PDCA, suggesting the preferential degradation of DHICA sub-units (Table 3), the highest amount being detected after treatment with ABTS/V and TEMPO/V (Table 3 entry 1 and 2). These results are similar to that obtained during the degradation of synthetic and natural eumelanin by chemical procedures [40,41,42,43,44]. The prevalence of the degradation of DHICA with respect to DHI is beneficial from the safety point of view. In fact, DHICA shows pro-oxidant activity that is responsible for the emergence of oxidative damage in the cell [53,54]. 

### 2.4. Structural and Morphological Characterization of Eumelanin after Treatment with LMCS

Field emission scanning electron microscopy (FE-SEM) of residual eumelanin after LMCS treatment with ABTS/V, TEMPO/V, Syr/V, As/V, and Va/V are reported in Figure 2A–F and compared with native eumelanin. Note that eumelanin is characterized by small nanoparticles, with an average diameter of 100–150 nm (Figure 2A) [55]. The residual eumelanin samples showed irregular morphology deprived of nanoparticles with a spherical shape (Figure 2B–F). These data confirm the efficacy of the polymer degradation [56]. Some residual nanoparticles were detected only in correspondence with the less efficient LMCS containing the As/V couple (Figure 2E). An analysis of previous samples using Attenuated Total Reflection Fourier Transform Infrared spectroscopy (ATR-FTIR) showed the increase of the intensity of O-H and N-H stretching in accordance with data previously reported for degraded eumelanin (Figure 3). In addition, a novel band at 2884 cm^−1^, 1513 cm^−1^, and 1139 cm^−1^ were observed and assigned to alkyl groups [4].

### 2.5. Preparation of LMCS Based Whitening Cream Formulations

Whitening cream formulations based on LMCS were prepared by using three different concentrations of the Va/V couple at a 1:1 ratio (1.6 μmol, 4.8 μmol, and 9.6 μmol), and three concentrations of laccase (0.79 U/mg, 1X; 2.37 U/mg, 3X; and 4.74 U/mg, 6X), taking care to separate laccase (chamber A) from the pair of redox mediators (chamber B) by an airless dispenser (Figure 4A). Chamber A contained dimethyl isosorbide (5% *w*/*w*) and sodium hyaluronate (1% *w*/*w*) and a variable amount of laccase in citrate buffer (94% *w*/*w*; pH 6.17, 0.1 M), whilst chamber B contained dimethyl isosorbide (5% *w*/*w*) and sodium hyaluronate (1% *w*/*w*) as well as the V/Va couple in citrate buffer (94% *w*/*w*; pH 6.17, 0.1 M). The DE of these formulations was evaluated by treating eumelanin (800 μg) dissolved in NaOH (200 μL, 0.5 M) with 1.0 mL of the content of chamber A, followed by the addition of 1.0 mL of the content of chamber B. The DE was calculated as described above and monitored by measuring the change in the absorbance of the sample (ʎ_MAX_ 540 nm) at 37 °C after 24 h. Irrespective of the experimental conditions, the whitening cream formulations showed appreciable capacity to degrade eumelanin, with the highest DE value being obtained in the case of the 6X sample (55%) (Figure 4). With the aim of further increasing the efficacy of eumelanin degradation, we performed a second run of treatment, with the selected 6X formulation obtaining an overall DE value of 78% (Figure 4). The stability of laccase in the 6X sample at 4 °C and 25 °C was evaluated by the standard ABTS assay at 420 nm [57]. As a general trend, the activity of laccase in the 6X formulation was fully retained at 4 °C, whilst the activity appreciably decreased at 25 °C.

## 3. Discussion

As reported in Table 1, the simple LMS did not efficiently degrade eumelanin, with the best result being obtained in the presence of V as a redox mediator (Table 1, entry 3). The effectiveness of the degradation increased in the presence of pairs of redox mediators (LMCS, Table 2). In this latter case, the Syr/V and Va/V couples showed an unexpected synergic effect (Table 2, entries 8 and 17) to yield DE values as high as 67% and 87%, respectively, after only 24 h of treatment. This effect occurred with redox mediators characterized by a high difference in redox potentials (Va/V: 0.77 V vs. 1.08 V and Syr/V: 0.66 V vs. 1.08 V), and it was not observed in the case of similar values (As/Av: 0.58 V vs. 0.57 V) [58,59,60]. The degradation efficiency was further improved by repeating the treatment for a second run under similar experimental conditions, affording the (almost) quantitative degradation of eumelanin in the presence of the Va/V couple (Table 2, entry 17). The high level of degradation was highlighted by the loss of the ordered and regular shape of eumelanin nanoparticles, as well as by the increase of the intensity of IR bands characteristic of the polymer degradation. In accordance with previous data, the degradation of eumelanin was associated with the typical increase in the intensity of OH and NH stretching and with the appearance of signals corresponding to alkyl groups [4]. The degradation pathway was evaluated by GC-MS determination of PTCA and PDCA (Table 3). The prevalence of PTCA suggested that the degradation of DHICA sub-units was more effective than their DHI counterpart, highlighting the beneficial effect of LMCS treatment in the selective removal of pro-oxidant residues in eumelanin. A similar trend was observed after the degradation of eumelanin with hydrogen peroxide and KMnO_4_ [42]. It is worth noting that optimal LMCS worked with high efficacy in the degradation of eumelanin, even when embedded in whitening cream formulations that resembled a standard commercial product. The use of an airless dispenser was essential to avoid the premature activation of laccase by redox mediators. In this latter case, the 6X whitening cream formulation showed a DE value similar to that of commercially available tyrosinase inhibitors such as 4-hexylresorcinol, vitamin B3 (niacinamide), and 1-amino-ethylphosphinic acid [61,62]. In addition, the storage temperature (4 °C) required for the stability of laccase is a normal feature of cosmeceutical products based on enzymatic formulations [63].

## 4. Materials and Methods

### 4.1. Materials

Melanin from *Sepia officinalis* was purchased from Sigma-Aldrich (St. Louis, MO, USA); laccase from *Trametes versicolor*, 2,2′-azino-bis(3-ethylbenzthiazoline-6-sulphonic acid) (ABTS), 2,2,6,6-Tetramethylpiperidin-1-yl-oxyl (TEMPO), Vanillin, 4-hydroxy-3-methoxybenzyl alcohol (Vanillyl alcohol), Acetosyringone, Syringaldehyde, and Acetovanillone were purchased from Sigma-Aldrich (St. Louis, MO, USA) and were used without any further purification. All experiments were performed in a citrate buffer pH 6.0 (0.1 M). Spectrophotometric measurements were taken with a Varian Cary 60 UV/Vis spectrophotometer equipped with a single-cell peltier thermostated cell holder. Spectrophotometric data were analyzed with the Cary Win UV software.

### 4.2. Degradation of Melanin by Laccase Mediator System (LMS)

As a general procedure, different amounts of redox mediators (0.4 μmol, 0.8 μmol, and 1.6 μmol) were added to eumelanin (800 μg) in an NaOH (200 μL, 0.5 M) and citrate buffer (2.4 mL; 0.1 M, pH 6) mixture, followed by the addition of laccase (0.79 U/mg) in buffer citrate (pH 6.0, 400 µL) at 37 °C for 4 h. The degradation of eumelanin was monitored by measuring the absorbance at the maximum melanin ʎ_MAX_ of 540 nm [46].

### 4.3. Degradation of Melanin by Laccase Mediator Cocktail System (LMCS)

As a general procedure, the selected redox mediator couple (1.6 μmol) was added to eumelanin (800 μg) in an NaOH (200 μL, 0.5 M) and citrate buffer (2.4 mL; 0.1 M, pH 6) mixture, followed by the addition of laccase (0.79 U/mg) in buffer citrate (pH 6.0, 400 µL) at 37 °C for 4 h. The degradation of eumelanin was monitored by measuring the absorbance at the maximum melanin ʎ_MAX_ of 540 nm [46].

### 4.4. Analysis of LMCS Oxidative Pathway by Gas Chromatography/Mass Spectrometry (GC/MS) Analysis

Mass spectrometry was performed with the use of a 450 GC-320 MS apparatus (VARIAN, Palo Alto, CA, USA) in comparison with commercial samples. Regarding the preparation of the sample, the residue was treated with pyridine, bis-trimethyl silyl trifluoro acetamide (BSTFA, 300μL) in the presence of *n*-dodecane (7.8 μmol) as an internal standard, under vigorous stirring conditions at room temperature for 2 h. The analyses were performed using a VF-5MS column and an isothermal temperature profile of 100 °C for 2 min, followed by a 10 °C/min temperature gradient to 280 °C for 25 min. The injector temperature was 280 °C. Chromatography-grade helium was used as the carrier gas, with a flow of 2.7 mL/min. The mass spectra were recorded with an electron beam of 70 eV.

### 4.5. FE-SEM Characterization

The field emission scanning electron microscopy (FE-SEM) results of the eumelanin samples were acquired by FE-SEM ZEISS GeminiSEM500 at 5 kV after a 20 mL drop (with deionized water) of the sample dispersion onto the specimen stubs, which were air-dried and coated with gold by sputtering with AGAR (Auto Sputter Coater). Before the observations, the samples received a deposition of chromium thin film (5 nm) by sputter coating with the use of a QUORUM Q 150T ES plus coater. 

### 4.6. Fourier Transform Infrared Spectra (FT-IR) 

Attenuated total reflection infrared (ATR-IR) spectroscopic analyses were performed at room temperature with a Perkin−Elmer Spectrum One spectrometer equipped with an ATR-IR cell. IR spectra were recorded by averaging 32 scans, with a resolution of 4 cm^−1^.

### 4.7. Activity Assay of Laccase 

To determine the activity parameters of laccase from *Trametes versicolor*, we used the 2,2′-azino-bis(3-ethylbenzothiazoline-6-sulfonic acid) diammonium salt (ABTS) assay. Laccase (0.79 U/mg) was added to citrate buffer (pH 6, 0.1 M, 3 mL) containing ABTS (0.58 μmol). The absorbance of the cationic radical ABTS^+^ was measured at 420 nm (molar extinction coefficient ε_420_ = 36 mM^−1^ cm^−1^). The increase of absorbance at 420 nm was measured at room temperature for 10 min. The enzyme activity was calculated, as described in the following equation, from the slope of absorbance versus the time curve: (2)$$U/\mathrm{mL}=\frac{Abs/min*Vtot}{\varepsilon^{mM}*Venz}$$

One unit of specific activity was defined as 1.0 mg of the enzyme consumed by 1.0 µmol of the substrate per minute. 

## 5. Conclusions

LMCS was an efficient tool for the degradation of eumelanin. We have demonstrated that the degradation significantly increased in the presence of the Syr/V and Va/V couples. In these cases, the degradation efficiency was 67% and 87%, respectively, after only 24 h of treatment. Worthy of note is the fact that the couples Syr/V and Va/V showed a synergic effect, probably due to their high difference in the value of the redox potential. The quantitative degradation of eumelanin was obtained by repeating the treatment with the selected LMCS in a second run. The prevalence of PTCA with respect to PDCA after the LMCS treatment suggests that the degradation of the DHICA sub-units was more effective than its DHI counterpart, favoring the removal of the more pro-oxidant residues in the starting material. It is also worth noting that LMCS based on the Va/V couple worked with high efficacy even when embedded in a whitening cream formulation that resembled a standard commercial product. The use of the airless dispenser was necessary to avoid the premature activation of laccase. The 6X whitening cream formulation performed similarly to commercially available tyrosinase inhibitors, reaching high stability at 4 °C [63]. Overall, these data suggest that LMCS is a potent tool for the development of alternative and sustainable whitening agents. 

## 6. Patents

Pending patent resulting from this work—Title: Procedure for the decolorization of eumelanin based on laccase cocktail mediators system and cosmetic and cosmeceutical application in the field of whitening cream. Number: 102022000008453; Deposition: 28 April 2022. 

## Figures and Tables

**Figure 1 ijms-23-06238-f001:**
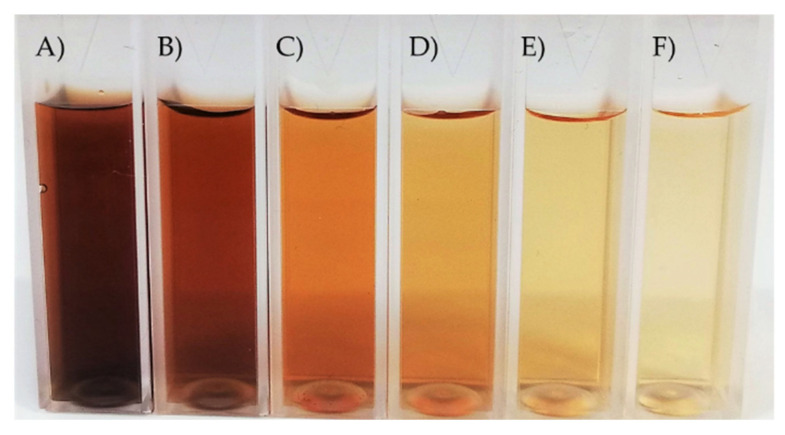
Comparison of the color intensity of eumelanin after treatment with LMCS at 24 h: (**A**) Sepia melanin as reference; (**B**) LMCS As/V; (**C**) LMCS TEMPO/V; (**D**) LMCS Syr/V; (**E**) LMCS ABTS/V; (**F**) LMCS V/Va.

**Figure 2 ijms-23-06238-f002:**
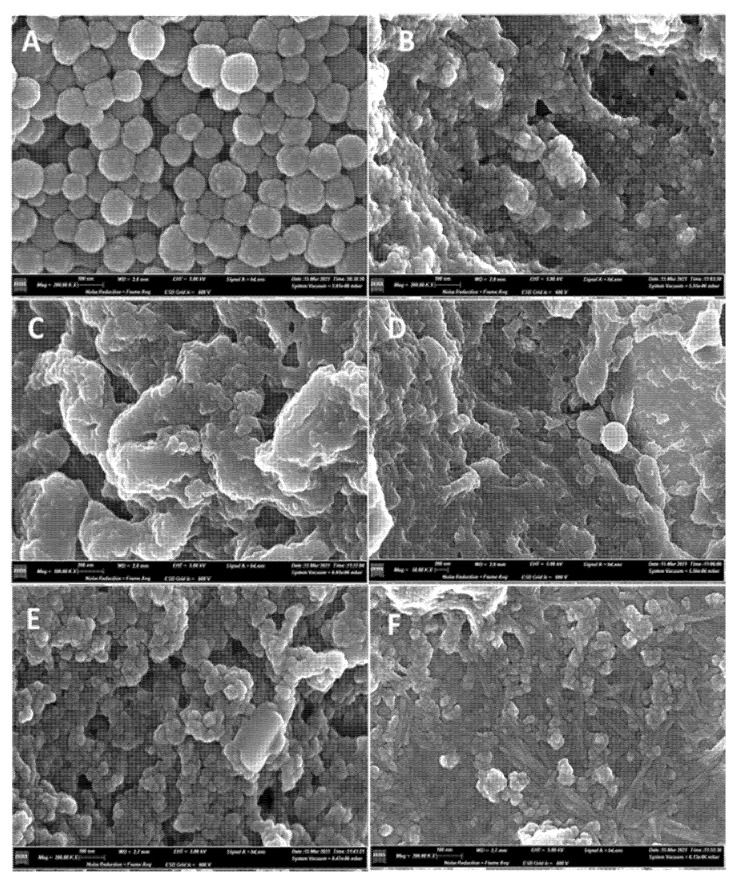
FE-SEM images of sepia melanin after LMCS treatments. Panel (**A**): native sepia melanin. Panel (**B**): sepia melanin after treatment with LMCS ABTS/V. Panel (**C**): sepia melanin after treatment with LMCS TEMPO/V. Panel (**D**): sepia melanin after treatment with LMCS Syr/V. Panel (**E**): sepia melanin after treatment with LMCS AS/V. Panel (**F**): sepia melanin after treatment with LMCS Va/V.

**Figure 3 ijms-23-06238-f003:**
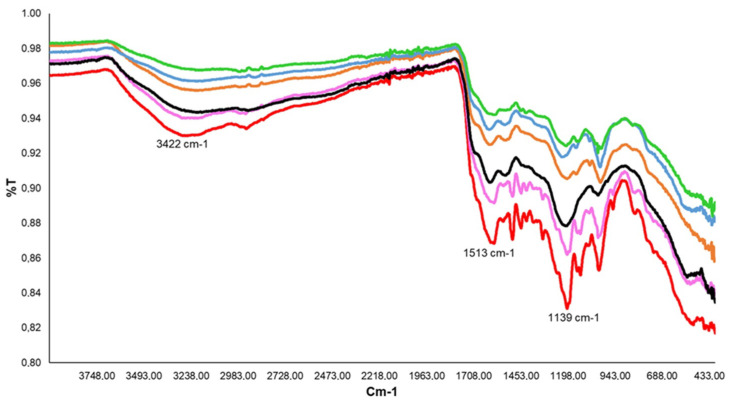
Attenuated total reflectance Fourier Transform Infrared spectroscopy (ATR-FTIR) of residual sepia melanin after LMCS treatment with ABTS/V (blue line), TEMPO/V (orange line), Syr/V (pink line), As/V (blue line), and Va/V (red line). Original Sepia melanin (green line) is reported as a reference.

**Figure 4 ijms-23-06238-f004:**
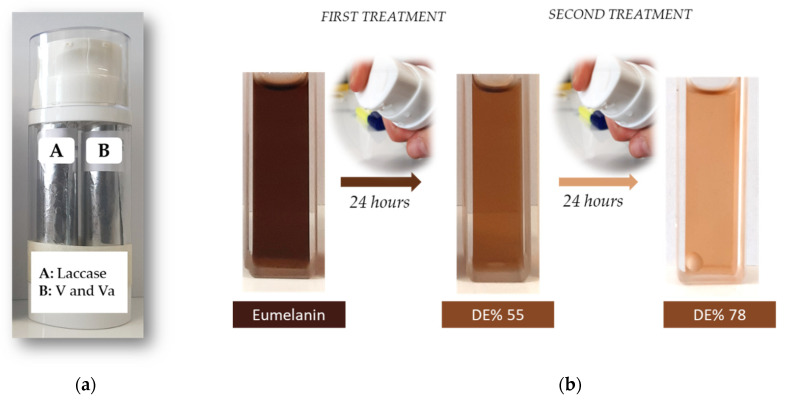
Panel (**a**): selected 6X formulation. Panel (**b**): application of the whitening cream 6X.

**Table 1 ijms-23-06238-t001:** Decolorization efficiency of eumelanin by LMS at different amounts of mediators (0.4, 0.8, and 1.6 μmol).

Entry	LMS	μmol	DE%
1	ABTS	(0.4) (0.8)(1.6)	(10) ^1^ (18) ^2^ (26) ^3^
2	TEMPO	(0.4) (0.8) (1.6)	(20) ^1^ (21) ^2^ (16) ^3^
3	V	(0.4) (0.8)(1.6)	(12) ^1^ (13) ^2^ (36) ^3^
4	Syr	(0.4) (0.8) (1.6)	(17) ^1^ (25) ^2^ (10) ^3^
5	As	(0.4) (0.8) (1.6)	(16) ^1^ (9) ^2^ (9) ^3^
6	Av	(0.4) (0.8) (1.6)	(8) ^1^ (14) ^2^ (16) ^3^
7	Va	(0.4) (0.8) (1.6)	(15) ^1^ (16) ^2^ (28) ^3^

^1^ Percentage of decolorization efficiency (DE%) calculated after 4 h of reaction at amount of mediators of 0.4 μmol. ^2^ DE% after 4 h at amount of mediators of 0.8 μmol. ^3^ DE% after 4 h at amount of mediators of 1.6 μmol.

**Table 2 ijms-23-06238-t002:** Eumelanin decolorization with LMCS in different molar concentration ratios and at different reaction times (4 h, 24 h, and 48 h).

Entry	LMCS	Molar Ratio ^1^	DE%(4 h) ^2^	DE%(24 h) ^2^	DE%(48 h) ^2^
1	ABTS/V	0.5:1	15	-	-
2	ABTS/V	1:1	50	70 (76) ^3^	67
3	ABTS/V	2:1	18	-	-
4	TEMPO/V	0.5:1	22	-	-
5	TEMPO/V	1:1	46	63 (76) ^3^	66
6	TEMPO/V	2:1	26	-	-
7	Syr/V	0.5:1	24	-	-
8	Syr/V	1:1	55	67 (78) ^3^	67
9	Syr/V	2:1	28	-	-
10	As/V	0.5:1	19	-	-
11	As/V	1:1	39	51 (61) ^3^	69
12	As/V	2:1	12	-	-
13	Av/V	0.5:1	12	-	-
14	Av/V	1:1	31	46	47
15	Av/V	2:1	14	-	-
16	Va/V	0.5:1	28	-	-
17	Va/V	1:1	67	87 (96) ^3^	71
18	Va/V	2:1	32	-	-
19	As/Av	0.5:1	15	-	-
20	As/Av	1:1	34	38	38
21	As/Av	2:1	10	-	-
22	Av/Syr	0.5:1	2	-	-
23	Av/Syr	1:1	3	9	6
24	Av/Syr	2:1	1	-	-
25	As/Syr	0.5:1	8	-	-
26	As/Syr	1:1	16	28	38
27	As/Syr	2:1	11	-	-

^1^ Ratio applied in order to reach a total amount of 1.6 μmol of redox mediators. ^2^ Percentage of decolorization efficiency calculated after 4 h, 24 h, and 48 h of reaction. ^3^ DE% calculated after the second cycle of LMCS, expressed as total value of DE% obtained after the two treatments (referred to the initial mg of melanin).

**Table 3 ijms-23-06238-t003:** Amount of PDCA and PTCA recovered after treatment of eumelanin with LMCS.

Entry	LMCS	PTCA (μg/mg_melanin_)	PDCA (μg/mg_melanin_)	Ratio PTCA/PDCA
1	ABTS/V	26.16	12.72	2.05
2	TEMPO/V	21.47	9.40	2.28
3	Syr/V	11.12	8.92	1.25
4	As/V	21.52	18.70	1.15
5	Va/V	6.11	5.14	1.18

## Data Availability

Not applicable.

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
