# Peer review of "Laccase Mediator Cocktail System as a Sustainable Skin Whitening Agent for Deep Eumelanin Decolorization"

_ijms, 2022, doi:10.3390/ijms23116238_

Round 1

Reviewer 1 Report

The research is well-designed and well-performed. And there are no comments regarding the conduct of the study.

However, there are few minor points recommended to improve the manuscript:

  • It is recommmedned, that the "Introduction" section be expanded and revised to inculde more detailed background on the topic.
  • There are few spell check errors which is recommended to be revised, such as the scientfic names of "Sepia Officinalis" and "Trametes Versicolor", which species name should initialize with small letter.
  • On page 3, first paragraph is copied by mistake from the MDPI tempelate and should be edited: "This section may be divided by subheadings. It should provide a concise and precise description of the experimental results, their interpretation, as well as the experimental conclusions that can be drawn." 

Author Response

Comments:

The research is well-designed and well-performed. And there are no comments regarding the conduct of the study.

We thanks reviewer 1 for the positive comment of the manuscript.

Q.1 It is recommended, that the "Introduction" section be expanded and revised to inculde more detailed background on the topic

A.1 We agree with the reviewer 1, the actual version of the Introduction was expanded and revised to include a larger background of the topic. The following sentence (and relative reference) were  introduced in the revised manuscript:

Page 2, Line 3-6,  “Examples of LMS degradation of melanin are reported, and the use of single redox mediator at a time, such as 1-hydroxybenzotriazole (HBT), 2,2'-azino-bis (3-ethylbenzothiazoline-6-sulfonic acid) (ABTS), acetosyringone, syringaldehyde, p-coumaric acid, vanillin, vanillic acid, vanillyl alcohol and acetovanillone, discussed in detail”.

Q.2 There are few spell check errors which is recommended to be revised, such as the scientfic names of "Sepia Officinalis" and "Trametes Versicolor", which species name should initialize with small letter.

A.2 Thanks, the names of species were corrected, accordingly.

Q.3 On page 3, first paragraph is copied by mistake from the MDPI template and should be edited: "This section may be divided by subheadings. It should provide a concise and precise description of the experimental results, their interpretation, as well as the experimental conclusions that can be drawn."

A.3 Thanks, the manuscript was corrected, accordingly.

Reviewer 2 Report

The authors have well-written manuscript regarding new sustainable treatment system for skin whitening for deep eumelanin decolorization. The article is a novel study, well designed and the claims are supported by the results. I hope the article will be interesting to scientific community and adds sufficient value to the literature regarding hyperpigmentation. I may pasting here the minor comments that the authors can follow to make it easier for the readers to follow it.

  1. English language should be improved.
  2. In methodology, 4.2 and 4.3 should be simplified.
  3. It is a good practice to demonstrate conclusion under separate heading instead that is drawn from the stated results.

Author Response

Comments:

The authors have well-written manuscript regarding new sustainable treatment system for skin whitening for deep eumelanin decolorization. The article is a novel study, well designed and the claims are supported by the results. I hope the article will be interesting to scientific community and adds sufficient value to the literature regarding hyperpigmentation. I may pasting here the minor comments that the authors can follow to make it easier for the readers to follow it.

We thanks reviewer 2 for the positive comment of the manuscript.

Q1. English language should be improved.

A1. The English language was revised, accordingly.

Q2. In methodology, 4.2 and 4.3 should be simplified.

A2. We agree with the reviewer 2, the 4.2 and 4.3 paragraph are now simplified.

Q3. It is a good practice to demonstrate conclusion under separate heading instead that is drawn from the stated results.

A3. We thanks reviewer 2, a novel conclusion section was added in the revised manuscript, accordingly.